# Osteopontin as a Regulator of Colorectal Cancer Progression and Its Clinical Applications

**DOI:** 10.3390/cancers13153793

**Published:** 2021-07-28

**Authors:** Katyana Amilca-Seba, Michèle Sabbah, Annette K. Larsen, Jérôme A. Denis

**Affiliations:** 1Cancer Biology and Therapeutics, Centre de Recherche Saint-Antoine (CRSA), 75012 Paris, France; katyana.amilca@inserm.fr (K.A.-S.); michele.sabbah@inserm.fr (M.S.); annette.larsen@sorbonne-universite.fr (A.K.L.); 2Institut National de la Santé et de la Recherche Médicale (INSERM) U938, 75012 Paris, France; 3Institut Universitaire de Cancérologie (IUC), Faculté de Médecine, Sorbonne Université, 75005 Paris, France; 4Centre National de la Recherche Scientifique (CNRS), 75016 Paris, France; 5Department of Endocrinology and Oncology Biochemistry, Pitié-Salpetrière Hospital, 75013 Paris, France

**Keywords:** osteopontin, colorectal cancer (CRC), biomarker, therapeutic target, metastasis

## Abstract

**Simple Summary:**

The mortality of colorectal cancer is principally related to metastatic disease at the time of diagnosis or to the growth of initially undetectable micro-metastasis. Current therapeutic strategies are efficient in patients with locally advanced cancer, but are rarely able to cure patients with metastatic disease. Therapeutic failure is mainly associated with drug resistance and an aggressive phenotype. The identification of new biomarkers for micro-metastasis and tumor progression remains an unmet clinical need that should allow for improved patient stratification for optimal treatment and may lead to the identification of novel therapeutic targets. Osteopontin (OPN), a multifunctional protein, has emerged as a potentially valuable biomarker in several cancer types. This review principally describes the molecular mechanisms of OPN that are associated with colorectal cancer (CRC) progression and metastasis, as well as the use of OPN as a clinical biomarker. This review identifies a role for OPN as a biomarker ready for extended clinical application and discusses its use as a therapeutic target.

**Abstract:**

A high expression of the phosphoprotein osteopontin (OPN) has been associated with cancer progression in several tumor types, including breast cancer, hepatocarcinoma, ovarian cancer, and colorectal cancer (CRC). Interestingly, OPN is overexpressed in CRC and is associated with a poor prognosis linked to invasion and metastasis. Here, we review the regulation and functions of OPN with an emphasis on CRC. We examine how epigenetic and genetic regulators interact with the key signaling pathways involved in this disease. Then, we describe the role of OPN in cancer progression, including proliferation, survival, migration, invasion, and angiogenesis. Furthermore, we outline the interest of using OPN as a clinical biomarker, and discuss if and how osteopontin can be implemented as a routine assay in clinical laboratories for monitoring CRC patients. Finally, we discuss the use of OPN an attractive, but challenging, therapeutic target.

## 1. Introduction

Colorectal cancer (CRC) is a major public health problem almost everywhere in the world. It represents the third most frequent neoplasm, with more than 1.9 million new cases and a mortality of around 935,000 cases each year worldwide [1]. In spite of the high mortality, CRC is a cancer with a good prognosis when diagnosed at an early stage and rapidly resected. For example, the five-year overall survival (OS) is around 91% for localized stages (stage I and II) and 70% for stages with loco-regional and lymph node invasion (stage III). However, approximately 25% of patients have detectable metastasis at the time of diagnosis (stage IV), with a five-year survival of around 11%. In addition, nearly 25% of patients initially diagnosed as non-metastatic will develop metastases later on, despite their surgical management, because of the presence of non-detectable micro-metastasis at the time of diagnosis. Consequently, the mortality attributable to this cancer type is mainly associated with the spread of the primary tumor to secondary sites, mostly in the liver and, less frequently, to the lung and peritoneum. Therefore, it is crucial to identify and understand the biological characteristics that render certain tumor cells aggressive, resulting in both local and systemic tumor dissemination. In particular, there is a clear clinical need for improved patient stratification in order to develop better strategies for therapeutic management. This may also facilitate the discovery of new potentially “druggable” targets and, possibly, to prevent and counteract tumor metastasis. Among the different clinicopathological parameters, blood-derived tumor markers have traditionally played a major role because they are minimally invasive, convenient for routine examination, and because the associated cost is low. The tumor markers are usually proteins that are either directly produced by the malignant cells or are produced by other cells in the tumor microenvironment in response to certain malignant conditions.

The carcinoembryonic antigen (CEA) and the carbohydrate 19-9 (CA19-9), discovered in 1965 and 1979, respectively, are still used in clinical practice today, especially for monitoring CRC. However, these parameters remain controversial for predicting the prognosis and surveillance of patients with CRC because of their lack of sensitivity and specificity. Moreover, except for *KRAS* mutations that are useful for predicting the response to EGFR-directed antibodies, very few clinically useful tumor markers have been discovered recently. In particular, no currently used markers are able to predict recurrence after surgery linked to initially undetectable micro-metastasis [2]. Thus, it is necessary to identify new molecular biomarkers to improve the management of CRC patients. Agrawal and co-workers have identified a protein named osteopontin (OPN) as the lead marker, which was most consistently associated with tumor progression using pooled sample expression profiling from a range of different clinical stages [3]. These findings were confirmed in a later study using tissue arrays to assess OPN protein levels in 350 tumors from 23 different body sites in comparison with 113 normal tissues. In particular, OPN was found to be elevated in tumors compared with the normal tissues, and to correlate significantly with tumor stage in several tumor types, including CRC [4]. Subsequent studies showed that OPN expression can be induced by the *RAS* oncogene, which is mutated in a large proportion of different cancers, especially in CRC, where *RAS* is mutated in at least 50% of cases [5]. In this review, we present a comprehensive review of the role of osteopontin in cancer progression with an emphasis on CRC [6,7]. We further discuss the clinical potential of using OPN as a biomarker for patient management and its use as a potential therapeutic target.

## 2. Regulation of Osteopontin

OPN has been cloned several times in different contexts. OPN was first described in 1979 by Richard Hynes and colleagues as a phosphoprotein specifically secreted by transformed cells lines [8]. An analysis of the OPN protein revealed the presence of a specific Arg-Gly-Asp (RGD) motif, which is known to serve for cell attachment by the heterodimer transmembrane receptors of the integrin family [9]. OPN was later described as a tumor-inducible gene [10]. In the same year, it was shown that activated T cells expressed high levels of osteopontin, which was thereore named the early T-lymphocyte activation-1 (Eta-1) protein [11]. OPN is encoded by the secreted phosphoprotein 1 (*SPP1*) gene localized on chromosome 4q21–25 within a gene cluster coding for the SIBLING protein family (Small Integrin Binding N-Linked Glycoprotein). This cluster also includes Dentin Matrix Protein1 (DMP1), Dentin Sialophosphoprotein (DSPP), Matrix Extracellular Phosphoglycoprotein (MEPE), and Bone Sialoprotein (BSP). All these proteins are components of the extracellular-matrix of bones and dentin, and play a role in the mineralization of these tissues. Unlike the other genes in this family, OPN is more widely expressed under both physiological and pathological situations. This includes numerous epithelial cells and their corresponding adenocarcinomas of the breast, lung, liver, stomach, ovarian, and colon, as well as melanoma and cells from normal and tumor-derived stroma, including cancer-associated fibroblasts (CAFs), tumor-associated macrophages (TAMs), infiltrating lymphocytes, endothelial cells, and smooth muscle cells.

### 2.1. Epigenetic Regulators

The osteogenic differentiation program is a major regulator of OPN expression. However, the broad expression of OPN, compared with the more restricted expression of the other SIBLING genes, suggests that the OPN promoter is also subject to gene-specific epigenetic regulation [12,13,14]. This process may include methylation of the CpG epigenetic marks in the vicinity of the OPN promoter, histone acetylation due to induction of histone acetyltransferases (HATs), and/or repression of histone deacetylases (HDACs), as well as chromatin remodelers and noncoding RNAs (Figure 1).

#### 2.1.1. Promoter Methylation, Histone Modifiers, and Chromatin Remodeling

DNA methylation is a process by which methyl groups are added to DNA. CRC typically displays both hyper- and hypo-methylation compared with normal colorectal tissue. When taking place in a gene promoter, DNA methylation typically acts to repress gene transcription without changing the DNA sequence. CRCs are mostly characterized by DNA hypo-methylation, which is associated with general genomic instability. However, they may also show gene-specific promoter methylation. In addition, a subset of CRCs display elevated methylation with a specific methylation pattern, termed the CpG island methylator phenotype (CIMP) [15].

The precise mechanisms controlling the methylation of the osteopontin promoter have not been fully elucidated. However, it was recently shown that the hypo-methylation of a CpG dinucleotide 647 base pair upstream of the transcription start site of *SPP1* was associated with shorter disease free survival in gastrointestinal stromal tumors (GIST). Moreover, treatment with 5-azacytidine, a DNA methyltransferase inhibitor, correlated with an increase of osteopontin expression in in vitro models [16]. A study in transgenic pigs reached the same conclusion, thereby reinforcing the importance of promoter methylation in the regulation of osteopontin expression [17].

Epigenetic regulation can also be mediated by the methylation and/or acetylation of histone tails, thereby altering the chromatin structure. Epigenetic regulators that regulate the transcriptional activity of OPN have mostly been investigated in the context of osteogenic differentiation [14]. However, a few publications have studied the epigenetic regulation of OPN in the context of cancer. Different histone deacetylase (HDAC) inhibitors have been tested and were shown to regulate OPN expression. Among them, suberoylanilide hydroxamic acid (SAHA), largazol, and valproic acid were all capable of up-regulating OPN. The up-regulation of OPN was associated with HDAC2 and/or HDAC8 inhibition, resulting in both increased acetylation of H3 (H3K9 and H3K14) and increased methylation of H3K4 in the OPN promoter [18,19,20,21,22]. Glucose may act as a potent inducer of histone acetylation and methylation, which in turn leads to upregulation of the OPN gene expression [23]. Additionally, the KDM6B histone demethylase promotes osteogenic commitment by removing the epigenetic H3K27me3 marks from the promoters of osteogenic genes, including OPN [24]. In contrast, the decrease of the OPN promoter activity in multipotent mesenchymal cells is associated with the enhancement of HDAC and H3K9 trimethylation [25].

#### 2.1.2. Non-Coding RNAs

Another type of epigenetic regulation is mediated by non-coding RNA, especially microRNA (miRNA), containing about 22 nucleotides that function in RNA silencing and the post-transcriptional regulation of gene expression. Osteopontin is negatively controlled by several miRNAs such as miR-181a and miR-96, which regulate the osteopontin-dependent metastatic functions in hepatocellular cell lines [26,27,28]. Other miRNAs such as miR-4262, miR-218-5p, and miR-466 negatively regulate OPN in osteosarcoma [28], triple negative breast cancer [29], and prostate cancer [30], respectively. Another publication has reported that miR-221 seems to accelerate OPN degradation. Interestingly, polyphenol epigallocatechin-3-gallate (EGCG), a component found in high concentrations in green tea that is able to suppress liver metastasis of human colorectal cancer, is able to downregulate OPN via up-regulating miR-221 [31]. Moreover, miR-221 is enriched in the plasma of patients with colorectal cancer and is correlated with p53 expression [32]. Finally, a panel of microRNAs termed osteomiRs is also able to regulate the osteogenic program, including OPN. The osteomiR panel is composed of the miR-30 family—miR-541 and miR-125b [33,34]. The deregulation of one or more of these regulators is likely involved in the aberrant expression of OPN in different cancer types, including colorectal cancer, and is believed to play a role in hepatic metastasis [35]. A recent publication suggests that OPN may also be a potential target of miR-146a-5p, which is known to promote the tumor phenotype [36]. However, this miRNA has been associated with both tumor progression [37] and the reduction of tumor growth and metastasis in mice [38]. Therefore, the role of miR-146a-5p in colorectal cancer is controversial, and more knowledge is needed to clarify the reported discrepancies. In conclusion, the epigenetic regulation of OPN in the context of colorectal cancer is still incompletely understood, but is probably important, as suggested by the prominent role of epigenetics in CRC [39,40].

### 2.2. Gene Expression and Signaling Pathways

The expression of OPN is also regulated by different transcription factors (TFs) and co-factors (Figure 1). The study of the human OPN promoter began in 1994 with an article by Hijiyia and coworkers [41], and a first comprehensive picture of the transcriptional regulation of the human osteopontin promoter emerged in 2000 [42]. The proximal promoter contains a region designated as RE-1, consisting of two cis-acting elements, RE-1a (−5 to −86) and RE-1b (−22 to −45), which act synergistically to regulate the activity of the OPN promoter. Gel shift assays demonstrated that RE-1a contains binding sites for transcription factors SP1, the glucocorticoid receptor, and the E-box-binding factors, whereas RE-1b contains a binding site for the octamer motif-binding protein (OCT-1/OCT-2). Since then, numerous cis-regulatory sequences have been identified in the OPN promoter, reflecting the complexity of the expression of this gene that depends on both the cellular context and the physiological and pathological situation. In this section, we will focus on the regulators and signaling pathways that are particularly relevant in colorectal cancer.

#### 2.2.1. Wnt/Wingless Pathway

The most common genetic alteration in colorectal cancer concerns the Wnt-signaling pathway [43], as more than nighty percent of human CRC tumors have molecular alteration in one or several genes in this pathway. Among them, missense mutations often appear in the APC tumor suppressor gene, leading to loss of function. In addition, there are other altered genes in this pathway, resulting in constitutive activation of the downstream signal. This leads to the disassembly of the AXIN/APC/GSK3β complex. Glycogene synthase kinase (GSK3β) is then no longer able to phosphorylate β-catenin, thereby preventing the proteosomal degradation of β-catenin. This leads to the stabilization of the non-phosphorylated form of β-catenin, which, in some cases, will undergo nuclear translocation. Nuclear β-catenin can bind the transcription factor T cell factor-4 (TCF4), which normally represses OPN expression directly at the promoter level. This results in the activation of the OPN expression, as well as of numerous other Wnt-signaling downstream genes [44]. In Rama 37 rat mammary epithelial cells, TCF4 suppresses OPN transcription and the metastatic phenotype [5], while the sequestration of the inhibitory TCF4 by specific DNA sequences containing the CAAAG motif is able to increase the osteopontin expression. On the other hand, high levels of TCF4 are inversely correlated with osteopontin expression in human breast carcinoma [45]. Among other aberrantly induced genes downstream of the nuclear β-catenin/TCF4 complex, some are direct OPN regulators that may participate in a positive regulation loop such as *c-MYC*, a major oncogene known to drive colorectal carcinogenesis [46,47]; the lymphoid enhancer-binding factor-1 (LEF1), which have prognostic value in CRC with respect to recurrence and metastasis [48]; and c-Jun, which is able to form by homo-dimers and hetero-dimers with c-FOS (called the Activator protein-1, AP-1), which is also implicated in CRC [49]. A separate study has demonstrated that the response of the OPN promoter to β-catenin and LEF-1 was considerably enhanced by the E26 transformation-specific (ETS) transcription factor family, including ETS-1/ETS-2, and ERM, which belongs to the polyoma enhancer activator 3 (PEA3) subfamily of ETS transcription proteins [50]. In particular, the ETS-1 transcription factor regulates OPN positively in the CT26 murine colorectal cancer model where it promotes liver metastasis [51]. The role of Wnt-signaling in the up-regulation of osteopontin has also been established in vivo using genetically modified mice that showed either constitutive activation of the Wnt-pathway (mutant APC^1638N^ mice) or not (pvillin-KRAS^G12V^) [44]. In particular, it was shown that APC gene mutations were necessary and sufficient for OPN upregulation in murine tumors both on the mRNA and the protein level. It was also shown that there is a strong correlation between OPN expression and nuclear β-catenin, with a highly significant relationship between co-expression of OPN and nuclear β-catenin and a poor prognosis. This finding was confirmed in a more recent work [52]. Taken together, these data indicate that OPN is a transcriptional target of aberrant Wnt-signaling in CRC, and that this might be a major mechanism in the upregulation of OPN.

#### 2.2.2. Bone Morphogen Protein/Transforming Growth Factor Pathway

The transforming growth factor-beta (TGFβ) superfamily plays a regulatory role both in the physiology of the normal colon and in the invasive CRC phenotype. The TGF superfamily ligands bind and signal through type II and type I serine/threonine kinase receptors (also called activin receptor-like kinases). These receptors include TGFBR2, TGFBR1, BMPR2, BMPR1A/1B, ACVR2A/2B, and ACVR1A/1B. In the canonical pathway, receptors interact at the cell surface upon ligand binding, leading to the trans-phosphorylation of type I receptors by the constitutively active type II receptors, resulting in downstream activation of pathway-specific receptor-associated SMAD proteins (R-SMAD proteins). The R-SMAD effectors then associate with SMAD4, a common SMAD for all TGF-β superfamily members, and this complex is translocated into the nucleus where it regulates the transcription of numerous target genes, including osteopontin. In colorectal cancer, nearly 80% of tumors display mutations in different members of the TGFβ family signaling pathway, including mutations in the genes for BMP, TGFβ receptors, and SMAD4. It is interesting to note that some mutations activate the pathway, whereas other lead to inactivation. In COS-1 cells, it has been shown that SMAD3 and SMAD4 exhibit different functions in the activation of OPN gene transcription. SMAD3 binds directly to the OPN promoter as a sequence-specific activator, while SMAD4 displaces the transcriptional repressor HOXA-9 through the formation of the SMAD4/HOX complex as part of the transcription response to TGF-beta stimulation [53]. Another relevant transcription factor for OPN regulation is the Runt-related transcription factor 2 (RUNX2), which is a downstream transcription factor of the SMAD-dependant TGFβ/BMP pathway and a direct activator of OPN capable of binding to a specific motif in its promoter [51]. Interestingly, it was reported that co-cultures of the CC531 rat colon adenocarcinoma cells with hepatocytes induce both OPN and RUNX2 [54]. RUNX2 is increasingly studied for its involvement in tumor progression, including colorectal cancer [55,56]. Other signaling pathways may also regulate RUNX2 expression, as suggested by a recently published study showing that liquiritigenin, a flavonoid extracted from the roots of *Glycyrrhiza uralensis*, remarkably reduced the expression of RUNX2 through inactivation of the phosphoinositide 3-kinase/protein kinase B (PI3K/AKT) pathway in HCT-116 CRC cells, and reduced both proliferation and invasion [57]. Otherwise, increasing evidence suggests that the TGFβ superfamily may also activate SMAD-independent (non-canonical) pathways and WNT-signaling pathways through signaling crosstalk, thereby mediating a complex downstream gene expression pattern.

#### 2.2.3. Vitamin D Receptor (VDR) Pathway

OPN expression is also transcriptionally regulated by the vitamin D receptor (VDR) pathway. Indeed, calcitriol (1α, 25-dihydroxyvitamin D3) binds to a high-specificity vitamin D response element (VDRE) in the OPN gene promoter [58]. This is coherent with the physiological role of osteopontin as a bone-related protein, as calcitriol is secreted by the proximal cells of the kidney in response to hypocalcaemia and hypophosphatemia. VDR is expressed in normal colon epithelial cells and by some colon cancer cells at variable levels [59]. An elevated VDR expression in CRC is associated with epithelial differentiation and favorable prognosis. However, in advanced cancers, VDR expression decreases or even disappears entirely, indicating that colon cancer cells can only express VDR as long as they maintain a certain level of differentiation [60,61]. Specific mutations may cause deletions, frame-shift mutations, premature stop codons, or splice site abnormalities that down-regulate VDR expression and/or binding activity during colon cancer progression. The attenuated VDR function is associated with ligand unresponsiveness and failure of therapy with vitamin D analogs. Otherwise, vitamin D3-deficiency has been proposed to play a role in colorectal cancer disease and progression. Indeed, diabetic rats exhibit vitamin D3 deficiency due to increased hepatic levels of IL-6 and OPN mRNA, which can be normalized by treatment with vitamin D3 [62].

#### 2.2.4. Nuclear Receptor Pathway

Different nuclear receptors have been reported to regulate OPN expression at the promoter level. This includes the transcription factor NR4A2 (nuclear receptor subfamily 4, group A, member 2), also named nuclear receptor related 1 protein (NURR1), which belongs to the steroid orphan nuclear receptor superfamily and is involved in inflammation through the regulation of prostaglandin E2 [63]. In addition, the estrogen-related receptor alpha (ERRα) has been implicated in the regulation of OPN expression in human colorectal cancer. Promoter analysis, inhibition of ERRα activity, and mutations of potential ERRα response elements in the proximal promoter of the human OPN gene showed that ERRα and its obligate co-activator, peroxisome proliferator-activated receptor γ co-activator-1α (PPARG coactivator 1α), positively control human OPN promoter activity. Chromatin immunoprecipitation experiments subsequently confirmed the in vivo occupancy of the OPN promoter by ERRα in HT-29 CRC cells, suggesting that OPN is a direct target of ERRα in colorectal cancer [64].

#### 2.2.5. Hedgehog Pathway

This pathway is relevant for the stem cell/stem cell-like phenotype in CRC, as the downstream targets of this signaling pathway include the stem cell-associated proteins LGR5, CD133, and CD44. The major downstream transcriptional factor for the Hh pathway is the GLIoblastoma zinc finger 1 protein (GLI-1), which is able to induce the osteopontin expression directly [65]. OPN expression is stimulated in the presence of different Hh ligands (Sonic, Indian, and Desert) and is inhibited in the presence of cyclopamine, a smoothened (SMO) inhibitor. Transcriptional silencing of GLI1 negatively affects OPN expression and compromises the ability of cancer cells to proliferate, migrate, and invade in vitro. The silencing of GLI1 also interferes with their ability to grow as xenografts and to metastasize in nude mice. All these phenotypic changes can be reversed by the re-expression of OPN in the GLI1-silenced cells, suggesting that OPN is a critical downstream effector of GLI1 signaling. In another study, the modulation of OPN by Hh ligands involved the Sex-determining region Y-box 9 (SOX9) transcription factors [66]. It was suggested that the balance between the transcription factors RUNX2 and SOX9, which regulate osteopontin expression in opposite directions, is a crucial element of this regulation.

### 2.3. Alternative Splicing, Alternative Translation, and Protein Localization

The human OPN pre-mRNA is subject to alternative splicing, which generates five confirmed isoforms termed OPNa, OPNb, and OPNc, as well as the newly identified OPN4 and OPN5. OPNa corresponds to the canonical full-length sequence (seven exons), OPNb has a deletion at exon 5 (∆58–71), OPNc has a deletion at exon 4 (∆31–57), while OPN4 (also termed OPNd) has a deletion in both exons 4 and 5. The OPN5 isoform displays an extra exon (eex), located between the canonical exons 3 and 4, which gives rise to an alternative translation start and a larger protein [67,68]. It is now well recognized that some OPN isoforms are cell context dependent, and might have different functions. For example, OPN-c is considered to be a breast cancer-associated isoform [69,70,71] that may play a critical role in the promotion of metastasis [72]. There is currently very limited information concerning isoform expression in colorectal cancer and the mechanisms that control alternative splicing. A recent study suggested that the serine/arginine-rich splicing factor 7 (SRF7) and different miRNA may play a role in renal cancer [73]. However, this should be taken with caution, as the splicing process is often tissue-specific. It has also been proposed that an increased expression of the splice variant OPN-c is associated with sensitivity to the anticancer drug 5-fluorouracil (5-FU) via the activation of the nuclear calcium signal. In agreement, 5-FU treatment of two CRC cell models, HCT-115 and HCT-8, was accompanied by an increased expression of OPN-c and, to a lesser degree, OPN-a and OPN-b, which was accompanied with increased apoptotic cell death and sensitivity to the 5-FU-dependent Ca^2+^ response. Other investigations have indicated that OPN-c expression is regulated by the binding of the epigenetic factor MeCP2, a calcium responsive regulator, on exon 4–5, and is modulated by its phosphorylation state [74].

In addition to alternative splicing, OPN can undergo alternative translation, which is important for the cellular location of OPN. Utilization of a 5′ canonical translation start site generates a protein that expresses an N-terminal signal sequence that targets OPN to secretory vesicles, thereby allowing OPN to act as a cytokine in the local tumor environment, as well as systemically. In contrast, use of a downstream start site generates a shorter protein without the N-terminal signal sequence, which is mainly localized in the cytoplasm [75] Intracellular OPN (OPNi) was initially observed in rat calvarial cells and described to be localized at the perinuclear membrane [76,77]. OPN can be cytoplasmic and exists as an integral component of a hyaluronan-CD44-ERM (ezrin–radixin–moesin) attachment complex that is involved in the migration of embryonic fibroblasts, activated macrophages, and metastatic cells, as demonstrated in OPN−/− mouse fibroblast [78]. Interestingly, OPN may also be nuclear. In HEK293 cells, nuclear osteopontin is able to bind polo-like kinase-1, a serine/threonine-protein kinase involved in cell cycle regulation [79]. Another study showed that MMP8 can cleave OPN and that the C-terminal region of OPN is localized in the nucleus [78].

### 2.4. Post-Translational Modifications

OPN is subject to different post-translational modifications including phosphorylation, glycosylation, sulfation, and proteolytic cleavage mediated by thrombin and different metalloproteinase (MMPs) [80,81,82]. Therefore, OPN can have a molecular weight ranging from 32.5 (full-length nascent protein) to 75 kDa due to the post-translational modifications. The phosphorylation of OPN is associated with integrin binding, cytokine secretion, and macrophage migration/activation [83,84]. Tumor-derived OPN is usually less phosphorylated than OPN derived from normal cells. For instance, RAS-transformed fibroblasts are known to produce OPN that is significantly under-phosphorylated [85]. Proteolytic cleavage of OPN by thrombin or matrix metalloproteinase (MMP) changes the biological functions of OPN in different diseases, including prostate cancer [86]. Thrombin is able to cleave OPN in order to unmask a cryptic epitope (SLAYGLR motif in mice) that allows for the recognition of integrins α9β1 and α4β1 [87,88]. Moreover, a modification in the C-terminal part significantly reduces the adhesion of cells to OPN via αVβ3-integrin. The inhibition of the αVβ3-integrin binding to OPN could be restored by proteolytic removal of the C terminus by thrombin and plasmin [89].

## 3. Role of Osteopontin in CRC

A considerable number of publications describe a role for OPN in cancer progression [7]. The different OPN isoforms seem to influence multiple steps during cancer progression in both tumor and tumor-associated cells via different signaling pathways (Figure 2). In this section, we review the biological functions of osteopontin on the metastatic cascade, with an emphasis on colorectal cancer.

### 3.1. Increased Proliferation and Cell Growth

Several studies have examined the influence of osteopontin on cell proliferation both in vivo and in vitro in colorectal cancer models. Huang and co-workers reported increased proliferation in HCT-116 human CRC cells exposed to OPN, which was associated with apoptosis inhibition. The addition of OPN-neutralizing antibodies was able to reverse both processes [46]. Inversely, the use of siRNAs targeting OPN in HT-29 CRC cells decreased the proliferation and altered the cell cycle. Flow cytometry analysis revealed a decreased fraction of cells in the S-phase of the cell cycle, which was correlated with a reduced expression of cyclin D1 protein [90]. These results indicate that OPN can promote cell proliferation through cell cycle modifications. The influence of osteopontin was also determined in tumor xenograft models. HCT-116 cells were treated or not treated with siRNA-OPN, followed by grafting in BALC/c nude mice. The results show that the tumor volumes were markedly smaller for the group with down-regulated OPN. A different study used OPN knock-down in CMT93CRC murine cells. The inhibition of the OPN expression was accompanied by decreased cell proliferation [46]. Another work tested the impact of OPN deficiency in a model of Min mice that were APC-deficient and developed polyps through Wnt-signaling activation. At 16 weeks, the number of small intestinal polyps in Min/OPN (+/−) and Min/OPN (−/−) mice was clearly less than that of Min/OPN (+/+) mice [91]. OPN knockout was also carried out in LS174T CRC cells followed by grafting in SCID mice. For comparison, the mice were also grafted with control cells that had been transfected with an empty vector. The results showed a slower growth of OPN knockout tumors compared with the empty transfection controls that still expressed OPN [92].

Interestingly, it has been proposed that chemotherapy may, in certain cases, mediate a paradoxical effect on cell growth. According to this model, the initial induction of tumor cell death could lead to a pro-inflammatory response that eventually will solicit cell growth [93]. This could, at least in part, be mediated by OPN, as OPN is able to act as a pro-inflammatory cytokine in response to cell debris. Whatever the mechanism, taken together, these studies clearly show that OPN is able to promote CRC proliferation in vitro as in vivo.

### 3.2. Inhibition of Cell Death

Many cell types are anchorage-dependent, meaning that they undergo apoptotic cell death when deprived of attachment to the basement membrane and the extracellular matrix (termed anoikosis). One study using HCT-116 CRC cells found that OPN can inhibit cell death signaling in anchorage-deprived cells through the regulation of both apoptosis and autophagy. This effect can be reversed by blocking the p38α MAPK, but not the ERK1/2, signaling pathway [94]. The OPN-mediated pro-survival effects are dependent on the expression of the variant exon 6 (V6)- or V7-containing CD44 receptor (CD44v), as demonstrated by the knockdown of the constitutively expressed V6-containing CD44 isoforms in HT-29 CRC cells. Generally, OPN-induced survival signaling has principally been attributed to the engagement of CD44v isoforms and the activation of an inside-out signaling cascade mediated by Src activation, leading to robust integrin activation and improved survival under environmental stress [93]. Other studies report that OPN use the phosphoinositol 3-kinase (PI3K) pathway to inhibit programmed death under anchorage-independent growth [95].

### 3.3. Migration and Invasion

Several studies have shown an effect of osteopontin on migration and invasion in CRC models both in vivo and in vitro. In one study, HCT-116 proliferation and migration were evaluated using wound healing and the trans-well (Boyden chamber) invasion assays. The study concluded that OPN increased both cell migration and invasion in a concentration-dependent manner. Moreover, neutralization of OPN with OPN-specific antibodies reduced both migration and invasion [96]. The precise mechanism involving OPN in CRC cell migration and invasion is not yet known. Among the different hypotheses, it was reported that recombinant human OPN was able to induce the expression of HOTAIR, a long, noncoding RNA strongly associated with the invasion and metastasis of cancer cells, in a time- and dose-dependent manner, mediated via the transcription factor interferon regulatory factor 1 (IRF1) [96]. Otherwise, invasion requires the expression of metalloproteases and urokinase-type plasminogen activator (uPA), a serine protease able to initiate proteolytic cascades. These processes could be regulated by osteopontin not only in tumor cells, but also by other cells in the tumor microenvironment. The signaling pathways involved in these processes are only partly identified, but are likely to involve the EGFR, the PI3K/AKT, and the αVβ3 integrin signaling pathways and, possibly, the epithelial-to-mesenchymal transition (EMT) (detailed further below).

### 3.4. Hypoxia and Neo-Angiogenesis

Angiogenesis is a complex process by which new capillary blood vessels are formed from the surrounding microvasculature. This can be further enhanced by the incorporation of circulating endothelial precursor cells. Neo-angiogenesis occurs when the tumors reach a certain size (usually 1–2 mm in diameter), which results in local hypoxia, lack of nutrients, and build-up of toxic metabolites. Neo-angiogenesis is a crucial step in cancer progression and represents one of the classical hallmarks of cancer [97]. Tumor angiogenesis depends predominantly on the VEGF (vascular endothelial cell growth factor) pathway. Accordingly, in CRC, anti-VEGF therapies such as bevacizumab and aflibercept have been developed and are routinely used for metastatic patients with RAS mutated tumors [98]. Hypoxia is considered the major physiological inducer of the angiogenetic process. Different reports have indicated that hypoxia can also regulate OPN expression. One study found that hypoxia led to an upregulation of the OPN protein expression, which was correlated with radio-resistance in different human CRC cell lines, including SW480, SW620, HT-29, and HCT-116 [99]. A different study investigated the effects of OPN knock-down on the angiogenesis of LoVo CRC cells. They found a decrease in the secretion of VEGF and the urokinase plasminogen activator (uPA) [100].

OPN may also stimulate angiogenesis directly by acting on the endothelial cells. The influence of OPN on sprouting, one of the earliest steps in the neo-angiogenic process, has been characterized [101]. During sprouting, there is activation of the endothelial cells, which is associated with the degradation of the underlying basement membrane and the surrounding extracellular matrix, followed by oriented migration toward the angiogenic signal. The migration of the endothelial cells is followed by a proliferative phase and then by differentiation into a capillary-like structure to form a microvascular network that can support blood flow. It has been reported that activated endothelial cells are detected next to OPN-expressing cells in CRC tissues. It has been speculated that the secreted OPN binds to αVβ3 integrin on the endothelial cells and induces HIF-1α expression through a PI3K/AKT/TSC2-mediated and mTORC1-dependent protein synthesis pathway, which in turn trans-activates the TCF12 gene expression. TCF12 then interacts with EZH2 and histone deacetylases to transcriptionally repress the VE-cadherin gene, thereby facilitating endothelial sprouting.

### 3.5. Induction of Epithelial to Mesenchymal Transition (EMT)

EMT (epithelial–mesenchymal transition) is a physiological process that is also associated with pathological conditions such as tumor progression. During this process, epithelial cells that normally adhere closely to each other thereby providing tissue coherence, lose their adhesion and become more migratory (that is they gain the ability to move in the sub-epithelial stromal tissue), as well as becoming more invasive (able to degrade the basement membrane and the extracellular matrix by secretion of metalloproteases, matrilysin, and other proteases). EMT has often been associated with increased resistance to different drugs, the acquisition of a stem cell-like phenotype, and the ability to induce angiogenesis and vascular mimicry. However, it is currently not clear if the EMT phenotype is causally related to drug resistance or rather is a parallel phenomenon that is favorized by the environmental stress cells undergo during drug exposure.

EMT is classified into different types. EMT type 1 is involved in physiological processes such as embryogenesis and allows the cells to fluctuate between a fully epithelial state and a fully mesenchymal state. EMT type 2 is associated with pathological fibrosis and is generally considered to be irreversible. EMT type 3 is linked to cancer progression and is regarded by some to be a partial reactivation of the developmental program [102,103]. It has been proposed by Thiery’s group that the level of EMT can be classified by a score, the so-called EMT score. The algorithm is based on gene expression, and ranges from −1 (for the most epithelial cells and tumors) to +1 (for the most mesenchymal cells and tumors). Importantly, CRC is mainly epithelial, suggesting that EMT in CRC is associated with a loss of epithelial features rather than the acquisition of a clear mesenchymal phenotype [102]. In 2015, Guiney et al. proposed four consensus subtypes (CSM) for CRC based on a wide-ranging genomic and transcriptomic characterization study combining six large studies [104]. This classification includes CSM1, microsatellite instability (MSI) and immune (14%); CSM2, canonical (37%); CSM3, metabolic (13%); and CSM4, mesenchymal (23%); 13% of cancers could not be classified into any of these groups and may represent a transition phenotype. Special attention was paid to group 4, the mesenchymal group, which was associated with the worst relapse-free and the worst overall survival. Importantly, group 4 patients also displayed the classical histopathological signs of severity including stromal infiltration and increased angiogenesis. For these reasons, the CSM4 group is believed to be strongly associated with the EMT process. In colorectal cancer, the main transcription factors regulating EMT include the Snail family (SNAI1 and SNAI2), the zinc finger E-box binding homeobox (ZEB) family, and the Twist bHLH transcription factor family, as well as the forkhead box family (FOXQ1, FOXC2, and FOXM1) and the prospero homeobox member PROX1 [105,106]. Recently, OPN has been proposed to be a master regulator of EMT in different cancer types [107]. In CRC models, it has been reported that OPN-overexpressing DLD1 cells may also display ectopic OPN secretion, overexpression of Snail and Twist1, and E-cadherin repression [108]. Additional work is required to better characterize the role of OPN on EMT induction, especially in the context of CRC.

### 3.6. Stemness and Drug-Resistance

Osteopontin is a potential modulator of “stemness”, the capacity of tumor cells to express factors and features classically associated with stem cells [109]. In colorectal cancer, different hypotheses have emerged to explain cancer initiation and auto-renewal of cells with stem cell-like properties that can be found in some tumors. One theory is that some intestinal normal stem cells (possibly the Lgr5 + cells) existing at the bottom of the crypts of Lieberkühn unit gain some driver mutations, contributing to initiating tumor growth followed by aberrant differentiation. A second hypothesis is that some more differentiated cells can partially dedifferentiate with the acquisition of stem cells properties. Finally, the expression of a more stem-cell like phenotype may be associated with the hypoxic tumor environment, which closely resembles that found during early embryogenesis, thereby promoting the expression of stem cell and developmental genes.

A recent paper reported that colorectal cells with increased stemness express CD44v6 (an established osteopontin receptor) [110]. It was shown that CD44v6 expressing cells are scarce in primary tumors, but that this subpopulation is favorized in the presence of OPN, or alternatively, in the presence of hepatocyte growth factor (HGF or stromal-derived factor 1α, SDF1) through the activation of the Wnt/β-catenin-pathway, thereby promoting migration and metastasis. Interestingly, CD44v6 (−) progenitor cells do not give rise to metastatic lesions, but, following cytokine treatment, acquire both the expression of CD44v6 and increased metastatic capacity. Importantly, phosphatidylinositol 3-kinase (PI3K) inhibition selectively kills CD44v6 tumor cells and reduces metastatic growth. The acquisition of stem cell-like properties is classically associated with drug resistance. However, in our experience, this might not necessarily be a general trait. We developed a panel of six CRC cell lines resistant to different chemotherapeutic agents universally used for the treatment of CRC, namely 5-fluorouracil (5-FU), oxaliplatin, and irinotecan (SN38) by chronic drug exposure [111]. Although chronic chemotherapeutic stress may lead to the acquisition of specific stem cell-like properties through the Wnt/wingless-pathway, the process was both heterogeneous, drug- and agent-dependent. In a recent paper, it was observed that DLD1 CRC cells overexpressing OPN also upregulated two classical “stem-cell associated” transcription factors, SOX2 and OCT4, and showed enhanced resistance to oxaliplatin. The increased resistance might be associated with the upregulation of members of the ATP-binding cassette sub-family, especially ABCG2 [112], which are established drug-transporters. Interestingly, patients with a high osteopontin expression often show increased oxaliplatin resistance in the clinic [108].

### 3.7. Immunity and Inflammation

Cancer progression and metastasis are often associated with inflammation. Osteopontin can promote both acute and chronic inflammation, and is able to modulate the function of several types of immune cells, including macrophages, dendritic cells, neutrophils, B and T lymphocytes, and natural killer cells in the tumor microenvironment [113]. Notably, osteopontin (OPN) can recruit monocytes that subsequently differentiate into tumor-associated macrophages (TAMs). Interestingly, the binding of OPN to TAM receptors such as integrins or CD44 is accompanied by the downregulation of nitric oxide, thereby decreasing the cytotoxic activity of the TAMs toward the surrounding tumor cells. Furthermore, OPN can be activated by matrix metalloproteinases (MMPs) into a smaller, pro-inflammatory molecules. Interestingly, cyclo-oxygenase 2 (COX2), a well-known inducible enzyme involved in inflammation, may play an important role in osteopontin production. For example, the presence of osteopontin was characterized in APC (Delta14/+) mice, which spontaneously generates polyps after treatment with parecoxib, an COX2 inhibitor [63]. Microarray analysis revealed that polyp-associated osteopontin was downregulated in COX-2 inhibited mice compared with the non-treated animals. The treatment also affected two other components involved in osteopontin regulation: the orphan nuclear receptor NR4A2 and the Wnt/beta-catenin-signaling pathway. Therefore, the down-regulation of osteopontin, probably through blockade of NR4A2 and Wnt-signaling, plays an important role in the antitumor activity of cyclooxygenase-2 inhibitors.

### 3.8. Survival, Protection, and Intravasation of Circulating Tumor Cells (CTCs)

The presence of circulating tumor cells (CTCs) is a potent independent prognostic factor for both overall survival (OS) and progression-free survival (PFS) in colorectal cancer [114]. It follows that biological processes that can protect circulating tumor cells from sheer and immune stress are likely to play a role in cancer progression. CTCs can interact with different circulating cells, including platelets and macrophages. It has been shown that macrophage secrete osteopontin, which may promote the survival of CTCs. As cell lines derived from CTCs including CTCs from CRC patients now exist, detailed functional studies are now feasible and should be able to provide clear answers to some of the outstanding questions concerning the CTC–macrophage interaction [115].

Another unsolved question is how CTCs can arrest and intravasate at locations distant from the primary tumors in order to establish micro-metastasis that may first become clinically relevant several years later. The existence of a “pre-metastatic niche” has been proposed, suggesting that the primary tumor can send molecular information to prepare a niche particularly favorable for the establishment of metastatic growth. One exciting hypothesis is that this might be mediated by exosomes that allow for the transport of RNA and proteins present in extracellular microvesicles that subsequently are able to reprogram recipient cells. Exosomes can carry different receptors including CD44v, which promotes motility, invasion, and metastatic growth in colorectal cancer cells [116], as well as integrins and osteopontin [117]. Future studies should be able to further clarify these questions [118].

### 3.9. Metastasis Formation

Colon cancer disseminates predominantly to the liver. It was found that OPN is highly expressed in metastatic hepatic lesions in CRC patients compared with the expression in the primary CRC tissue and its adjacent normal mucosa. The roles of OPN in the liver also include the promotion of hepatocellular cancer and are believed to facilitate the establishment and growth of CRC metastasis [119,120]. The molecular processes are not fully understood but may involve the mesenchymal–epithelial transition (MET), the reversal of EMT, thereby allowing the disseminated tumor cells that display a more mesenchymal phenotype to revert to a more epithelial phenotype during the later stages of metastatic dissemination. Interestingly, OPN also plays a role in decidualization, a process by which the fibroblast-like endometrial stromal cells differentiate into polygonal epithelial-like cells [121]. Recently, the mesenchymal–epithelial transition was connected with the presence of intracellular/nuclear OPN (iOPN) [122]. As already mentioned, the isoform, and thus the location, of OPN varies as a function of the tumor type. Type I cells express only secreted osteopontin (sOPN), type II express sOPN and iOPN, and type III only express iOPN. Under normal culture conditions, human colorectal cell lines such as 293T, LS174T, LoVo, and Colo25 secrete little or no OPN into the media, whereas the excretion can reach considerable levels for other cell lines (Caco2). Immunofluorescence reveals a nuclear staining pattern of OPN in many colorectal cancer cell lines. It has been suggested that the downregulation of OPNi is associated with EMT (decrease of E-CADH and increase of αSMA, N-CADH, VIM, ZEB1, and ZEB2), whereas the forced overexpression of iOPN is associated with the inverse process (MET), including the increase of E-CADH expression and decrease of vimentin.

It has also been suggested that the vascular endothelial growth factor (VEGF) can re-localizate OPN in the nuclei both in vitro and in vivo via activation of phospholipase Cγ (PLCγ) and Protein Kinase C (PKC), which then phosphorylate the intracellular OPN on a specific serine residue. Then, nuclear phospho-Ser OPN can recognize HIF2α, thereby stimulating the hypoxia-responsive element of the AKT1 promoter. Consequently, downstream miRNA (such as miR429) may negatively regulate SEC23A, a protein transport protein involved in the secretion of OPN, and prevent EMT by the negative regulation of ZEB1 and ZEB2. Although this mechanism is attractive, it remains to be seen whether this is a general process that is relevant within the context of CRC. One mechanism in favor of a process linking hypoxia and OPN is provided by a recent paper [123]. It was reported that the restoration of oxygen in metastatic colon cancer through Myo-Inositoltrispyrophosphate (ITPP), a nontoxic, anti-hypoxic molecule that recently has shown significant benefits in experimental cancer models, particularly when combined with standard chemotherapy, inhibits tumor spread and is accompanied by a reduction in malignant serum markers including osteopontin, leading to markedly improved animal survival.

## 4. Potential Applications of OPN as a Biomarker

An important number of articles suggest that OPN could be useful as a prognostic and diagnostic biomarker in both tissues and blood in different types of cancer. In the following section, we provide a short review about the clinical utility of osteopontin in the context of colorectal cancer.

### 4.1. Screening and Early Detection

Screening programs for the early diagnosis of CRC have been implemented in most industrialized countries. In particular, colonoscopy and (immunochemical) fecal occult blood testing have been useful to decrease CRC-associated mortality [124]. However, around 50% of patients are only diagnosed at an advanced stage (III or IV), which has a major pejorative impact on their prognostics. Progress is still needed to develop a test that is high performing, low-cost, and widely acceptable for the general public. Generally, blood-based tests are preferred for screening. In 2010, Roche diagnostics conducted a marker identification program [125]. They selected a panel of six markers consisting of CEA, ferritin, seprase, anti-p53 autoantibody, CYFRA21-1, and osteopontin. After testing of 1027 different serum samples at 95% specificity, osteopontin only showed a sensibility of 30.2% (lower than CEA at 43.9%). However, the univariate analysis of OPN (as measured by the AUC on ROC curve) is 0.73 (0.69–0.78), which is similar to CEA. Combinations of the two markers gave a sensitivity of 69.6% for a specificity of 95%, which is comparable to the performance of fecal immunochemical testing (FIT). These findings were later confirmed in a retrospectively clinical study using 1660 samples collected prospectively and analyzed retrospectively [126]. The osteopontin levels were significantly higher in the serum from CRC patients than in the serum from healthy controls (HC), with a mean of 44 ng/mL (13–306.5) versus 17.3 ng/mL (2–63), respectively. However, osteopontin serum levels can also be elevated in benign colorectal cancer, with a mean of 26.7 (5.3–147). The AUC value was 0.62 (0.52–0.71) in this study, in agreement with the data of the Roche cohort. In a small independent study with 106 patients, the diagnostic performance of a multi-parametric assay using 11 markers including OPN was evaluated [127]. All of the tested parameters were higher in CRC patients versus normal subjects. However, most of parameters including OPN were also elevated in benign colorectal disease. The results were concordant with other studies, and rather disappointingly suggest that the best combination is the one already known, which associates CEA and CA19-9. In a recent work, a blood test called Cancer SEEK^®^ was developed aiming to detect eight common human cancer types. The test detects eight circulating protein biomarkers (including OPN) as well as some tumor-specific mutations in circulating DNA, and seems to be particularly useful for discriminating cancer patients from healthy subjects. In a study of 1000 patients previously diagnosed with cancer and 850 healthy control individuals, CancerSEEK^®^ detected cancer with a sensitivity of 69 to 98% (depending on the cancer type) and 99% specificity [128]. Altogether, these studies systematically found that osteopontin was present in higher levels in the serum of CRC patients compared with normal subjects. However, osteopontin alone was not superior to the tumor marker combination currently used (CEA and CA19-9). These data are in agreement with the close association of OPN with inflammatory features, which are prevalent both in benign colon disease and in CRC (although generally at higher serum levels).

### 4.2. Prognostic Value in CRC Patients

In 2015, a comprehensive meta-analysis was carried out on the prognosis and clinic-pathological parameters in patients with colorectal cancer [129]; 1698 patients from 15 studies were included in the meta-analysis. The pooled data showed that a high OPN expression was significantly associated with high tumor grades (OR = 2.24, 95% CI 1.55–3.23), lymph node metastasis (OR = 2.36, 95% CI 1.71–3.26), and distant metastasis (OR = 2.38, 95% CI 1.01–5.60). Moreover, a high OPN expression was significantly associated with the two-year (HR 1.97, 95% CI 1.30–3.00), three-year (HR 1.82, 95% CI 1.24–2.68), and five-year (HR 1.53, 95% CI 1.28–1.82) survival rates and with overall survival (OS, HR 1.70, 95% CI 1.12–2.60), respectively. However, this meta-analysis was based on heterogeneous data, as the osteopontin levels were determined either in tissues or blood using a variety of different methodologies including reverse transcription polymerase chain reaction (RT-PCR), immunohistochemistry (IHC), and enzyme-linked immunosorbent assay (ELISA).

### 4.3. Evaluation of Recurrence Rate after Surgery

A better estimation of the risk of recurrence after surgery represents an urgent clinical need required to optimize the therapeutic management and to determine which patients should be offered adjuvant chemotherapy. The results from a small cohort of 79 CRC patients indicated that a high post-operative (but not pre-operative) plasma levels of OPN (>153.02 ng/mL) were correlated with the development of metastasis after curative resection, as well as with shortened disease-free survival [108]. In contrast, neither pre-operative nor post-operative CEA were correlated with disease free-survival (DFS). Further studies on larger cohorts are clearly needed to validate these findings in the post-operative situation. For example, retrospective studies comparing the metastatic risk in the adjuvant chemotherapy arm versus no treatment group for patients with Stage II CRC according to their serum or plasma concentration of OPN would be of clear interest.

### 4.4. Potential Immune Marker

CRC is currently stratified according to the TNM (tumor, lymph node, and metastasis) classification. Recently, a new classification system named immunoscore or TNM-i has been proposed, which is based on the densities of CD3 + or the CD8 + immune cells present in the tumor center and at the invasive edge. This strategy specifically aims to predict the prognosis of patients with MSI (microsatellite instability) status (15–20% of all CRC patients), which typically is able to elicit a strong immune response. So far, immunotherapy with immune checkpoint inhibitors has only been shown to be effective in CRC patients with MSI status. The identification of a biologically relevant gene expression pattern that both correlates with resistance to immunotherapy and the activation of cytotoxic T cells would be useful to improve the immunotherapy of CRC. It is well established that tumor-infiltrating immune cells exert their functions through reciprocal communication with neoplastic cells via different mediators. OPN is one of the relevant mediators in stromal cells of the tumor microenvironment, and is also secreted by lymphocytes [130,131]. Knock-out of OPN in CT26 mice in combination with immunotherapy decreased the number of tumor nodules. Furthermore, the combination of anti-OPN monoclonal antibodies and anti-PD1 is more efficient in reducing tumors nodules than anti-PD1 immunotherapy alone [132].These data suggest that OPN can compensate for PD-L1 to promote immune escape of the tumor, thereby rendering the tumor cells resistant to immunotherapy. Furthermore, OPN may serve as a predictive biomarker for CRC patients with MSI receiving immunotherapy. The targeting of OPN may increase the effectiveness of immunotherapy in CRC.

## 5. Potential Applications of OPN as a Therapeutic Target

Osteopontin can be considered as a promising therapeutic target because of its implication in multiple processes associated with CRC progression [133,134]. Different therapeutic strategies have been proposed to target OPN in vivo, including (i) ribozyme cleavage or hybridisation with either antisense oligonucleotides or, most recently, nuclear acid aptamer [135]; (ii) blocking antibodies or synthetic RGD peptides [136,137]; and (iii) small-molecule inhibitors such as parecoxib. Parecoxinib is a cyclo-oxygenase-2 inhibitor that reduces OPN expression, most likely through blockade of the NR4A2 and Wnt signaling pathways [63]. Despite encouraging initial results with the different OPN inhibitors, the clinical utility remains to be confirmed. The factors that currently limit the use of OPN-directed strategies include (i) the rapid turn-over rate of OPN and (ii) the presence of splicing variants, as well as numerous post-translational modifications of OPN for which the biological roles and importance are still poorly understood. Finally, osteopontin-directed strategies will not only target the tumor cells, but also the cells in the tumor environment, which could both be an advantage and, possibly, a disadvantage, depending on the contribution of the implicated tumor-associated cells to tumor progression.

## 6. Concluding Remarks and Perspectives

OPN has emerged as a multifactorial protein that serves as a master regulator of cancer progression. In CRC, OPN is of clear interest, but additional data are needed in order to use OPN as a marker for routine use in clinical laboratories. First, OPN exists in different isoforms that may have specific functions in different tumor types. This underlines the need to develop specific antibodies and to determine which motif is the best target. Next, we need to define a meaningful cut-off value and to determine if serum or plasma should be used. Finally, the development of assays specifically developed for the standard analytical systems routinely used in clinical laboratories would likely improve the sensitivity and specificity needed for clinical applications.

In general, it appears that the best assay to identify patients with elevated risk for metastatic disease will be a multi-parametric assay. We believe osteopontin should be included in such tests, which, once optimized, could be extremely useful for patient stratification. We conclude that OPN is probably not a useful biomarker by itself within the context of CRC, as it can be up-regulated in other pathological situations. However, OPN might be viewed as a general marker of cancer progression, which would be very useful in combination with other biomarkers in order to guide patient stratification and therapeutic strategies, which remain a real clinical need for the better treatment of CRC patients. Finally, OPN is an attractive therapeutic target because of its multiple roles in the promotion of tumor aggressiveness. However, further research will be needed to be able to target the relevant osteopontin isoforms in order to limit off-targets and potential side effects.

## Figures and Tables

**Figure 1 cancers-13-03793-f001:**
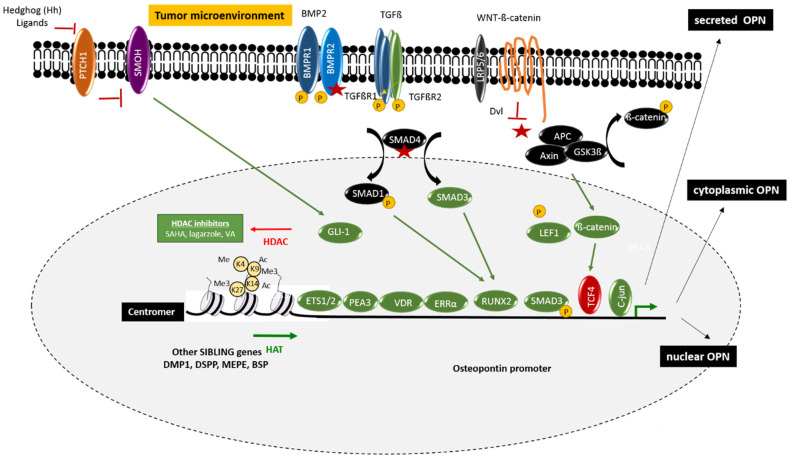
Epigenetic and genetic regulation of the OPN promoter in CRC. The regulation of the osteopontin expression is complex and involves both genetic and epigenetic mechanisms such as promoter methylation, as well as different protein complexes acting on the histones and thereby regulating the compaction of chromatin within the SIBLING gene cluster. The osteopontin promoter also contains numerous binding sites for transcription factors that act in coordination to regulate the transcriptional activity. Some are known to be connected to important CRC signaling pathways, particularly the Wnt pathway, as well as the TGFβ pathway and the Hedgehog pathway. The pre-messenger is also subject to alternative splicing generating different isoforms and contains an alternative initiation site leading to the formation of a shorter protein. Osteopontin may be intracellular (nucleus, cytoplasm, and secretory vesicle) or extracellular following secretion. Osteopontin is believed to play a role in the liver both as a matrix protein and as a cytokine mediating signaling between tumor cells, as well as between cells in the tumor environment, such as cancer-associated fibroblasts (CAFs) and tumor-associated macrophages (TAMs), which also secrete osteopontin.

**Figure 2 cancers-13-03793-f002:**
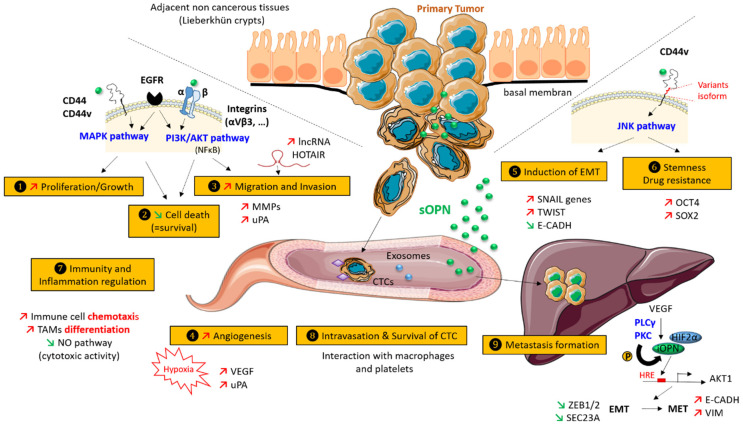
Role of OPN in CRC progression. OPN is involved in cancer progression by regulating various aspects of tumor progression and metastasis, including (1) increased proliferation and cell growth, (2) decreased cell death, (3) increased migration and invasion, (4) neo-angiogenesis, (5) promotion of the epithelial-mesenchymal transition (EMT), (6) increased stemness and drug resistance, (7) modulation of immunity and inflammation both at the primary and metastatic sites, (8) intravasation and survival of circulating tumor cells (CTC), and (9) promotion of liver metastasis. sOPN—secreted OPN; iOPN—intranuclear OPN; MET—mesenchymal to epithelial transition; HRE—HIF response element.

## Data Availability

Not applicable.

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
