# Peer review of "Osteopontin as a Regulator of Colorectal Cancer Progression and Its Clinical Applications"

_cancers, 2021, doi:10.3390/cancers13153793_

Round 1
Reviewer 1 Report
In this study, Amilca-Seba et al. summarize the functional roles of osteopontin (OPN) in colorectal cancer (CRC). This review is well-organized covering most of topics that readers possibly want to know. The authors describe from regulation and expression of OPN and then discuss the roles of OPN in CRC. This review also covers the potentials of OPN as a biomarker. This is an excellent review and I have only a few minor comments for this manuscript: (1) It would be great and easier to understand if there are associated pathways/genes/proteins shown in Figure 2 (e.g., Snail for EMT and VEGF for angiogenesis); (2) The authors mention the potentials of OPN as a biomarker, but how about the potentials as a therapeutic target? Anti-OPN antibodies or small molecules inhibiting OPN? Enhancing regulation against OPN expression by overexpression of genes or miRNAs? Difficulties or limitations in current therapeutic approaches targeting OPN?
Author Response
We thank the reviewer for the careful review of our manuscript on the regulation and roles of osteopontin in the progression of colorectal cancer. We appreciate the positive comments and useful suggestions. We have now modified figure 2 in order to associate the signaling pathways, genes and functions of osteopontin. In addition, we have added a small paragraph concerning the potential role of osteopontin as a therapeutic target. We hope the manuscript is now ready for publication and thank you again for your time and effort.
Reviewer 2 Report
The review describes the molecular mechanisms of action of the multifunctional protein Osteopontin (OPN) and focus on its association with colorectal cancer progression and metastasis and on its potential use as a clinical biomarker.
The review is well articulated, all major fuction of OPN are clearly described and documented with a rich and updated bibliography.
Author Response
We thank you for taking the time to review our manuscript regarding the regulation and role of osteopontin in the progression of colorectal cancer.
We are delighted that you enjoyed our manuscript and we hope this will also be true for the readers of « Cancers ». Thanks again.